# Measurement report: Exploring the $NH_3$ behaviors at urban and suburban Beijing:
## Comparison and implications
Ziru Lan[a], Weili Lin[a], Weiwei Pu[b], Zhiqiang Ma[b,c]
[a]College of Life and Environmental Sciences, Minzu University of China, Beijing 100081;
[b]Environmental Meteorological Forecast Center of Beijing-Tianjin-Hebei, Beijing, 100089, China;
[c]Beijing Shangdianzi Regional Atmosphere Watch Station, Beijing, 101507, China
Correspondence: Weili Lin (linwl@muc.edu.cn)
**ABSTRACT**
Ammonia ($NH_3$) plays an important role in particulate matter formation, and hence its atmospheric level
is relevant to human health and climate change. Due to different relative distributions of $NH_3$ sources,
the concentrations of atmospheric $NH_3$ may behave differently in urban and rural areas. However, few
parallel long-term observations of $NH_3$ to reveal the different behaviors of the $NH_3$ concentrations at the
urban and rural sites in a same region. In this study, online ammonia analyzers were used to continuously
observe atmospheric $NH_3$ concentrations at an urban site and a suburban site in Beijing from January 13,
2018, to January 13, 2019. The observed mixing ratio of $NH_3$ averaged 21 ± 14 ppb (range: 1.6–133 ppb)
at the urban site and 22 ± 15 ppb (range: 0.8–199 ppb) at the suburban site. The $NH_3$ mixing ratios at the
urban and suburban sites exhibited similar seasonal variations, with high values in summer and spring
and low values in autumn and winter. The hourly mean $NH_3$ mixing ratios at the urban site were highly
correlated ($R = 0.849$, $P < 0.01$) with those at the suburban site. However, the average diurnal variations
in the $NH_3$ mixing ratios at the urban and suburban sites differed significantly, which implies the different
contributions of $NH_3$ sources and sinks at the urban and suburban sites. In addition to the emission
sources, meteorological factors were closely related to the changes in the $NH_3$ concentrations. For the
same temperature (relative humidity) at the urban and suburban sites, the $NH_3$ mixing ratios increased
with relative humidity (temperature). Relative humidity was the factor with the strongest influence on
the $NH_3$ mixing ratio in different seasons at the two sites. The relationships between the $NH_3$
concentrations and temperature (relative humidity) varied from season to season and showed differences
between the urban and suburban sites. The reasons for the different relationships need to be investigated
in future studies. Higher wind speed mainly from the northwest sector lowered the $NH_3$ mixing ratios at
both sites. Similar with other primary pollutants in Beijing, the $NH_3$ mixing ratios were high under
impacts of air masses from the south sector.
**Keywords:** $NH_3$; variations; simultaneous observation

## 1. Introduction

Ammonia ($NH_3$) is the most abundant alkaline trace gas in the atmosphere (Meng et al., 2017). An excessive $NH_3$ concentration directly harms the ecosystem; causes water eutrophication and soil acidification; and leads to forest soil erosion, biodiversity reduction, and carbon uptake variations (Pearson and Stewart, 1993; Reay et al., 2008; Van Breemen et al., 1983; Erisman et al., 2007). $NH_3$ can react with acidic gases to form ammonium salts, which might significantly influence the mass concentration and composition of particulate matter (Wu et al., 2009). As major components of fine particle, ammonium salts contribute largely to the scattering of solar radiation and hence influence climate change (Charlson et al., 1991). Therefore, atmospheric $NH_3$ is one of the key species relevant to human health, ecosystem and climate change.

After the implementation of policies such as the *12th Five-Year Plan for the Key Regional Air Pollution Prevention and Control in Key Regions* (Ministry of Ecology and Environment of the People's Republic of China, 2012) and the *Air Pollution Prevention and Control Action Plan* (General Office of the State Council, PRC, 2013), China, especially the capital city Beijing, has been effectively controlling the emissions of sulfur dioxide ($SO_2$) and nitrogen oxide ($NO_x$), which are key precursors of fine particles. However, the pollution caused by fine particles is still serious (Krotkov et al., 2016; UN Environment, 2019), particularly in winter in the North China Plain, where excess $NH_3$ promote the haze formation through heterogeneous reactions (Ge et al., 2019). Studies have indicated that when the $SO_2$ and $NO_x$ concentrations are reduced to a certain extent, reducing $NH_3$ emissions is the most economical and effective method to decrease the $PM_{2.5}$ concentration (Pinder et al., 2008). In China, the main anthropogenic sources of $NH_3$ are livestock and poultry feces (54%) and fertilizer volatilization (33%) (Huang et al., 2012). Moreover, the atmospheric $NH_3$ concentration in China has increased with the

expansion of agricultural activities, control of $SO_2$ and $NO_x$, and increase in temperature (Warner et al.,
2017). This increase in the $NH_3$ concentration might weaken the effectiveness of $SO_2$ and $NO_x$ emission
control in reducing $PM_{2.5}$ pollution (Fu et al., 2017).

The North China Plain is a region with high $NH_3$ emission (Zhang et al., 2017), and Beijing has one

of the highest $NH_3$ concentrations in the world (Chang et al., 2016b; Pan et al., 2018). Compared with
studies on pollutants such as $SO_2$ and $NO_x$, considerably fewer studies have been conducted on the $NH_3$
concentration in Beijing. Chang et al. (2016a) collected gaseous $NH_3$ samples during the 2014 APEC
summit (October 18 to November 29, 2014) in the Beijing urban area and concluded that the overall
contributions of traffic, garbage, livestock, and fertilizers to the $NH_3$ concentration were 20.4%, 25.9%,
24.0%, and 29.7%, respectively. According the data from Huang et al (2012), the $NH_3$ emissions in
Beijing were from livestock and poultry farming (34.55%), nitrogen-fixing plants (33.57%), fertilizer
use (13.06%), household garbage treatment (8.29%), traffic emissions (5.20%), industrial emissions
(0.14%), biomass combustion (0.42%), and agricultural soil (0.84%). Zhang (2016) measured the $NH_3$
concentrations in urban and rural areas of Beijing from January to July 2014 and found that $NH_3$
concentration in urban areas was approximately 65% higher than that in rural areas. Meng et al. (2011)
reported that the highest $NH_3$ concentration in Beijing occurred in summer and the lowest one occurred
in winter, and their results indicated traffic to be a significant source of $NH_3$ in urban areas. Zhang et al.
(2018) reported the vertical variability of $NH_3$ in urban Beijing based on one-year passive sampling in
2016/2017 and concluded that local sources such as traffic emissions were important contributors to
urban $NH_3$. Meng *et al*. (2020) investigated the significant increase in winter $NH_3$ and its contribution to
the increasing nitrate in $PM_{2.5}$ from 2009 to 2016, and they also concluded that vehicles exhaust was an
important contributor to $NH_3$ in urban Beijing in winter.
Currently, NH₃ is not included in the routine environmental monitoring operation in China. Research
data on NH₃ monitoring, particularly on the synchronous observations of $NH_3$ concentrations with a high
temporal resolution in urban and suburban areas, are relatively scarce. In this study, high-time-resolution
observations of $NH_3$ were obtained simultaneously at an urban site and a suburban site in Beijing. The
variation characteristics and influencing factors of the $NH_3$ concentration were analyzed with
meteorological data to provide a scientific basis for $NH_3$ pollution control in Beijing.
**2. Materials and methods**
*2.1. Measurement sites*
From January 2018 to January 2019, continuous and simultaneous observations of atmospheric $NH_3$
were conducted at an urban site and a suburban site in Beijing. The urban site was located on the roof of
the Science and Technology Building of Minzu University of China (39.95°N, 116.32°E, altitude: 102
m) and the suburban site was in the Changping Meteorological Station (40°13′N, 116°13′E, altitude: 77
m). The suburban site is in the NW direction relative to the urban site and the shortest distance between
these two sites is approximately 32 km (Figure 1). More farm land and glass land are around the suburban
site than the urban site.

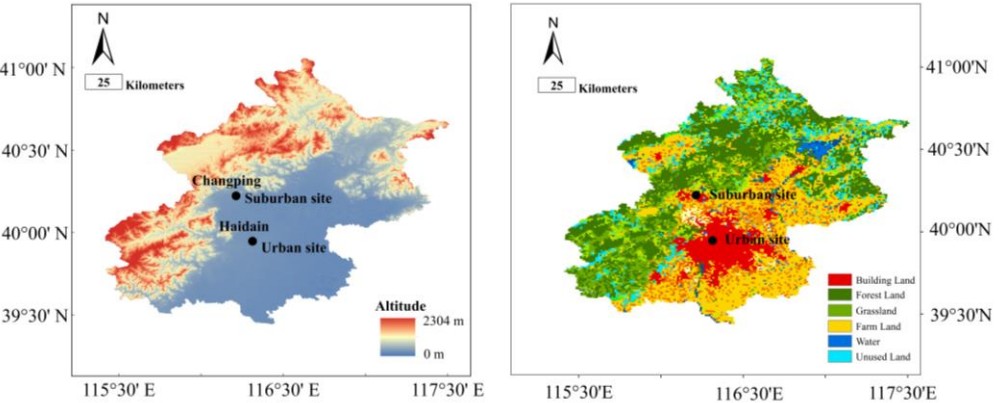


**Fig. 1.** Location of the observation sites, the topography, and land use of Beijing city.

*2.2. Measurements and data acquisition*

NH$_3$ concentrations were measured using two NH$_3$ analyzers (Ammonia Analyzer-Economical, Los Gatos Research Inc., USA), which have the minimum detection limit of <0.2 ppb and the maximum drift of 0.2 ppb/24hrs. The NH$_3$ analyzers were deployed in air-conditioning rooms. These analyzers use off-axis integrated cavity output spectroscopy (OA-ICOS) technology, which is a fourth-generation cavity-enhanced absorption technique, to simultaneously measure NH$_3$ and water vapor (H$_2$O) in the atmosphere. The incident laser beam of the OA-ICOS technology deviates from the optical axis, which differs from the traditional coaxial incidence mode. The axial incidence mode of the OA-ICOS technology can increase the optical path, stimulate additional high-order transverse modes, effectively suppress the noise of the cavity mode, reduce the cross interferences and errors due to contaminants existing in the cavity, and improve the detection sensitivity (Baer *et al*., 2002; Baer *et al*., 2012). The analyzer method is a quasi-absolute measurement, which theoretically does not require calibration. However, to ensure the comparability of the obtained data with other monitoring data, NH$_3$ standard gas (Beijing AP BAIF Gases Industry Co., Ltd.) was used for comparison measurement before the observation. The recorded concentrations were revised with respect to the reference NH$_3$ concentration in the standard gas mixture.

Ambient air was drained at >0.4 L/min through Teflon lines (1/4'OD) from a manifold. The lengths of the Teflon lines were designed as short as possible (less than 2 m from the manifold). Particulate matters were filtered by Teflon membranes with a pore size less than 5 μm. Since NH$_3$ easily "sticks" to surfaces (like inside walls of tubes), heated sample lines were suggested by many measurement studies. However, according our test (Fig. S1) in the lab, when heating (70℃) was on, there did have a peak lasting 5–6 min minutes and then deceasing to the normal levels in ambient air, which means a new balancing process has been established in less than 10 min. This suggests that heating is not necessarily

a solution for NH$_3$ sticking. Keeping the relatively stable balance between adsorption and desorption of
NH$_3$ in the sampling system is important. When tested using air of different humidity, only very sharply
changes of humidity obviously influenced and changed the balance, and a new balance needed tens of
minutes to reestablished (Fig. S2). Under the normal weather conditions, humidity changes in a relatively
smoothing way unless a quickly changing weather system, like rain, is approaching. The minute-level
data were converted into hourly averages in the data analysis process and the hourly resolution can
smooth the effect to some extent caused by variations in humidity and temperature during the observation.
The balancing idea was also used to carry out multi-point calibrations on NH$_3$ analyzers (Fig. S3).
A high mixing ratio (e.g. 400 ppb or higher) of NH$_3$ mixing gases were firstly produced by a dynamic
diluter and measured by the NH$_3$ analyzers overnight. After the signals reached the stable level, other
lower span values were switched in turn. At each span point, the measurement time was lasting at least
40 minutes or longer. Then a linear regression function was obtained with R$^2$ higher than 0.999.
Nowadays, NH$_3$ in compressed gas cylinder is also trustworthy, as confirmed by the comparison with the
NH$_3$ standard in a permeation tube (Fig. S4).
Totally, 7645 and 8342 valid hourly mean observations were obtained for the urban (Haidian) and
suburban (Changping) sites, respectively. In addition, the urban and suburban meteorological data
(temperature, relative humidity, wind direction, and wind speed) during the sampling period were
obtained from the Haidian Meteorological Observation Station and Changping Meteorological Station,
respectively.
**3. Results and discussion**
*3.1. Overall variations in the NH$_3$ mixing ratios*
Fig. 2 displays the time-series variations in the NH$_3$ mixing ratios, temperatures, and relative
humidity at the urban and suburban sites in Beijing. At the urban site, the mean ± 1σ, median, maximum,
and minimum values of the hourly average $NH_3$ mixing ratio during the observation period were 21 ± 14
ppb, 17 ppb, 133 ppb and 1.6 ppb, respectively. At the suburban site, the corresponding values were 22
± 15 ppb, 18 ppb, 199 ppb, and 0.8 ppb, respectively. The annual average and range of the $NH_3$ mixing
ratio at the suburban site were marginally higher than those at the urban site. The characteristics of the
weekly smoothed data indicate that the $NH_3$ variations and temperature/humidity fluctuations at the two
sites were practically consistent, which suggests that both sites were under the influence of similar
weather systems. The hourly mean $NH_3$ concentrations at the urban site were significantly correlated ($R$
= 0.849, $P < 0.01$) with those at the suburban site.

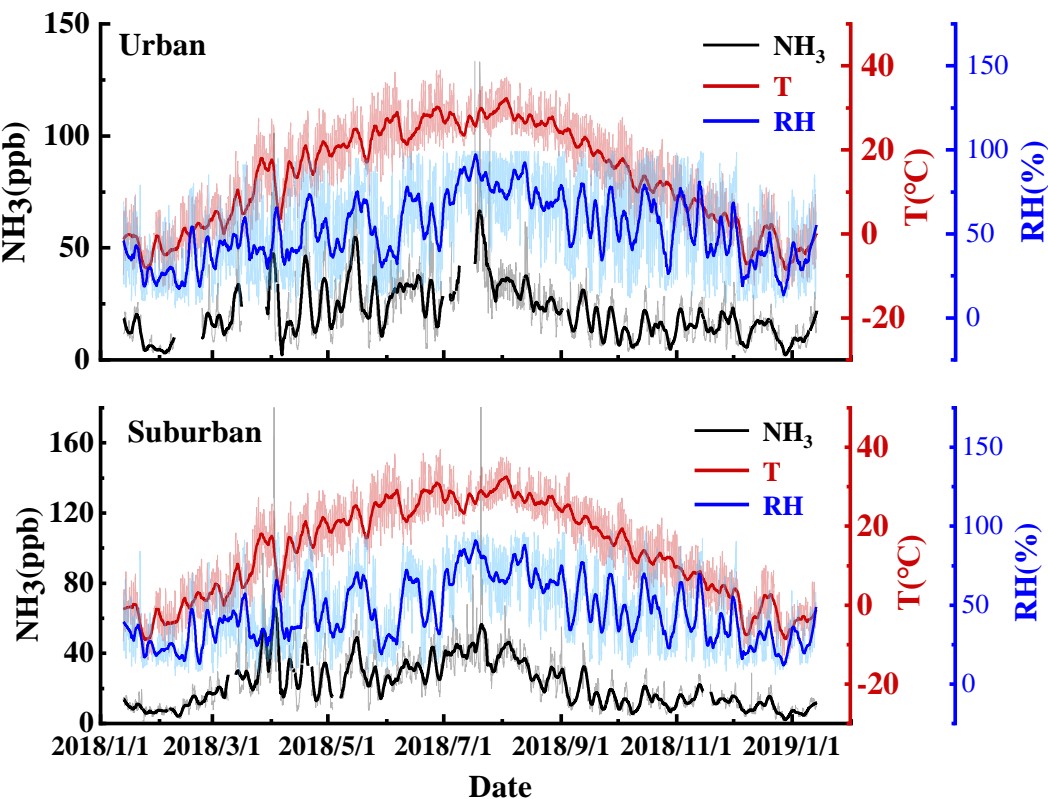

**Fig. 2.** Temporal variations in the hourly average $NH_3$ mixing ratios, temperatures ($T$) and relative humidity ($RH$) at the urban and suburban
stations in Beijing. Continuous thick lines were smoothed with 168 points (7 days) by using the Savitzky–Golay method.

Table 1 shows the comparison of atmospheric $NH_3$ concentrations (ppb) observed in different areas.

Meng et al. (2011) obtained an average NH$_3$ mixing ratio of 22.8 ± 16.3 ppb for the period 2008-2010 in

Beijing urban area, which is very close to our result (21 ± 14 ppb) for 2018-2019. Therefore, the annual

average NH$_3$ mixing ratio in urban Beijing did not change significantly from 2008 to 2019. Moreover,

results from this study and Meng et al. (2011) indicate that the NH$_3$ concentrations at the urban and

suburban sites were higher than those in the background areas. The observed NH$_3$ concentrations in

Beijing were higher than those in northwest China (Meng et al., 2010) and the Yangtze River Delta region

(Chang et al., 2019). The average annual NH$_3$ concentration in the urban area of Shanghai, a mega city

in the Southeast China (31° N), was approximately 50% lower than that in urban Beijing. This might be

related to the fact that the North China Plain, in which Beijing is located, is one of the most intensive

agricultural production regions in China. The differences in the soil properties of Beijing and Shanghai

may be another reason because the loss of soil NH$_3$ can increase with an increase in the soil pH (Ju et al.,

2009). Shanghai and its surrounding areas are dominated by acidic soil of paddy fields (Zhao et al., 2009),

whereas Beijing is dominated by the alkaline soils of dry land (Wei et al., 2013). In addition, the climate

in Beijing is much drier than in Shanghai so that less atmospheric NH$_3$ in Beijing can be removed than

in Shanghai by wet deposition.

**Table 1.** Comparison of the atmospheric NH$_3$ concentrations (ppb) observed in different areas.

| Period | Location | Methodology | Types | Concentration | Reference |
|---|---|---|---|---|---|
| 2018.01-2019.01 | Beijing, CN | Online monitor | Urban | 20.8±13.7 | This study |
| | | | Suburban | 21.9±14.9 | |
| 2008.02-2010.07 | Beijing, CN | Passive sampler | Urban | 22.8±16.3 | Meng et al., 2011 |
| 2007.01-2010.07 | | | Background | 10.2±10.8 | |
| 2014.5-2015.6 | Shanghai, CN | Passive sampler | Urban | 7.8 | Chang et al. 2019 |
| | | | Suburban | 6.8 | |
| 2006.04-2007.04 | Xi'an, CN | Passive sampler | Urban | 18.6 | Cao et al. 2009 |
| | | | Suburban | 20.3 | |
| 2017.12-2018.2 | Hebei, CN | Online monitor | Rural | 16.7±19.7 | He et al. 2020 |
| 2008 | Qinghai, CN | Passive sampler | Rural | 4.1±2.2 | Meng et al. 2010 |

| 2003.7-2011.9 | Toronto, CA | Passive sampler | Urban | 2.3-3.0 | Hu et al. 2014 |
| | | | Rural | 0.1-4 | |
| 2016.4-2017.10 | New York, US | Active and passive system | Urban | 2.2-3.2 | Zhou et al. 2019 |
| | | | Rural | 0.6-0.8 | |
| 2017.12 | Tokyo, JP | semi-continuous microflow analytical system | Urban | 4.1 | Osada et al. 2019 |
| 2013.1-2015.12 | Delhi, IN | Automatic analyzer | Urban | 53.4±14.9 | Saraswati et al., 2019 |
| 2012.10-2013.9 | Jaunpur, IN | Glass flask sampling | Suburban | 51.6±22.8 | Singh and Kulshrestha, 2014 |
| 2008.1-2009.2 | Bamako, MLI | Passive sampler | Urban | 46.7 | Adon et al., 2016 |
| 2006.3-2017.4 | Edmonton, CA | Online monitor | Urban | 2.4±0.6 | Yao et al., 2016 |
| 2010.9-2011.8 | Seoul, KR | Online monitor | Urban | 10.9±4.25 | Phan et al., 2013 |
| 2004.3-2004.7 | Munster, DE | Wet denuder | Urban | 5.2 | Vogt et al., 2005 |


Table 1 also shows observational results of atmospheric $NH_3$ from some other countries. The $NH_3$
mixing ratios in the United States (Edgerton et al., 2007; Nowak et al., 2006; Zhou et al. 2019), Scotland
(Burkhardt et al., 1998), Canada (Hu et al., 2014), Japan (Osada et al., 2019), and Germany (Vogt et al.,
2005) were 0.23–13 ppb, 1.6–2.3 ppb, 0.1–4 ppb, 4.1 ppb, and 5.2 ppb, respectively. These values are
considerably lower than those in Beijing. However, Delhi, India (Saraswati et al., 2019), exhibited higher
$NH_3$ mixing ratio (53.4±14.9 ppb) than Beijing did. This result might be attributed to the well-developed
livestock breeding activities in Delhi. This comparison indicates that in the decade before 2019, the $NH_3$
concentration in Beijing did not change considerably, but it is of the highest in big cities in China and
much higher than those observed in developed countries in America, Europe and Asia.
*3.2. Seasonal variations*
Fig. 3 displays the monthly statistics for the $NH_3$ mixing ratios at the urban and suburban sites in
Beijing. The seasonal variations in the $NH_3$ mixing ratios were very similar at the urban and suburban
sites, with higher mixing ratios in the spring and summer and low ones in the autumn and winter. The
daily mean concentrations fluctuated considerably in the spring, particularly in April. The highest mean
$NH_3$ concentrations at the urban and suburban sites were 42± 17 ppb and 42 ± 8.2 ppb, respectively. Both
occurred in July, when the $NH_3$ concentrations fluctuated considerably as well. On average, the seasonal
$NH_3$ mixing ratios at the urban and suburban sites can be arranged as follows: summer > spring > autumn >
winter. The main grain crops in the rural area of Beijing are corn and wheat. Corn is categorized as spring
corn and summer corn, which are sown in April and June, respectively. Usually, a large amount of base
fertilizer is applied when planting corn and the topdressing after 2 months. Wheat is sown from
September to October, and the topdressing is applied in the following spring. The volatilization of
nitrogen fertilizers can cause an increase in atmospheric $NH_3$ mixing ratios and its fluctuations in
fertilization seasons (Zhang et al., 2016). In addition, the high temperature in summer should also be
responsible to the high $NH_3$ mixing ratios in this season. An increase in the temperature can increase the
biological activity and thus enhance the $NH_3$ production and emission. High temperature is also
conducive for the volatilization of the urea and diammonium phosphate applied to crops. Moreover, the
equilibrium among ammonium nitrate particles, gaseous $NH_3$, and nitric acid is transferred to the gas
phase at high temperature, which increases the $NH_3$ concentration (Behera et al., 2013). Sewage treatment,
household garbage, golf courses, and human excreta are crucial $NH_3$ sources that are easily neglected
(Pu et al., 2020). The relatively low $NH_3$ concentrations in the autumn and winter might be caused by the
decrease in $NH_3$ emission in the soil and vegetation, the decrease in the $NH_4NO_3$ decomposition capacity
at low temperatures, and the reduced human activities caused by a large floating population returning to
their native locations outside Beijing during the Spring Festival (Liao et al., 2014). In the spring and
summer, the $NH_3$ mixing ratios at the suburban site were higher than those at the urban site, which might
be related to the higher agricultural activity around the suburban site. In the autumn and winter, the $NH_3$
mixing ratios at the urban site were marginally higher than those at the suburban site. In the autumn and
winter seasons, the influences of agricultural activities on the $NH_3$ concentration were weakened,
whereas the influences of other sources (such as traffic sources) were enhanced. According to Wang et
al. (2019), the traffic $NH_3$ emission per unit area in Haidian (urban site) was three times higher than that
in Changping (suburban site). This difference in traffic source emissions might have resulted in higher
$NH_3$ concentrations at the urban site than at the suburban site in the autumn and winter.

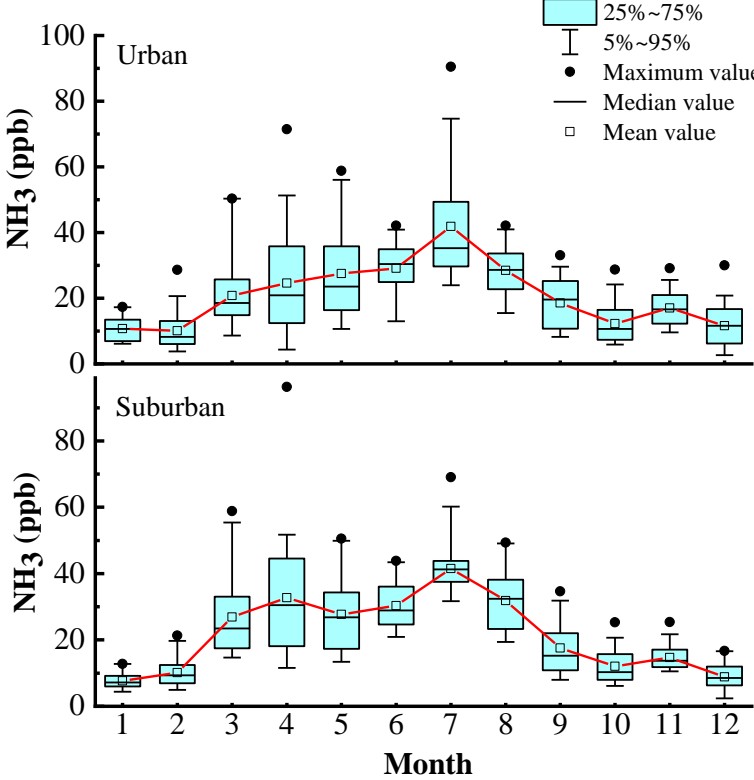


**Fig. 3.** Monthly statistical variation in the $NH_3$ mixing ratios at the urban and suburban sites in Beijing.

**Table 2.** $NH_3$ mixing ratios (ppb) measured at the urban and suburban sites in Beijing.

| Site | Time period | Mean | Standard deviation | Minimum | Median | Maximum |
|------|-------------|------|--------------------|---------|--------|---------|
|       | Annual | 21 | 14 | 1.6 | 17 | 133 |
|       | Spring | 25 | 16 | 1.9 | 21 | 101 |
| Urban | Summer | 32 | 12 | 5.0 | 30 | 133 |
|       | Autumn | 16 | 7.5 | 3.8 | 15 | 41 |

| | | | | | |
|---|---|---|---|---|---|
| | Winter | 11 | 6.7 | 1.6 | 9.9 | 42 |
| | Annual | 22 | 15 | 0.8 | 18 | 198 |
| | Spring | 29 | 16 | 6.8 | 26 | 180 |
| Suburban | Summer | 35 | 12 | 12.1 | 33 | 199 |
| | Autumn | 15 | 6.8 | 4.1 | 13 | 55 |
| | Winter | 9.2 | 4.5 | 0.8 | 8.4 | 29 |


*3.3. Diurnal variations*

Figure 4 displays the average diurnal variations in the NH$_3$ and H$_2$O mixing ratios in different seasons at the urban and suburban sites in Beijing. Ambient NH$_3$ exhibited different diurnal behaviors in different seasons.

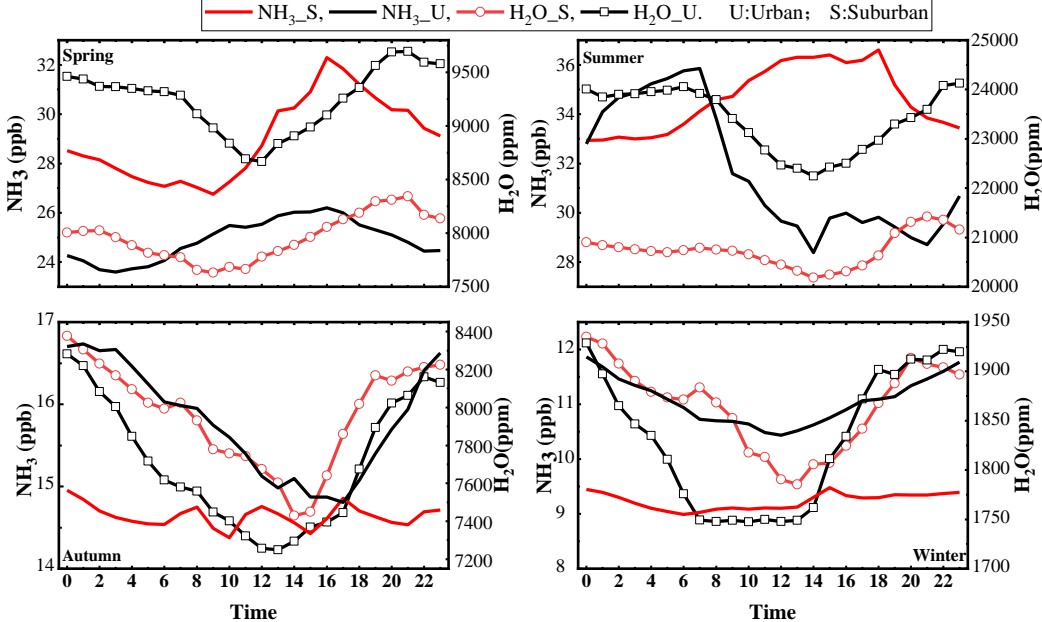


**Fig. 4.** Average diurnal variations in the NH$_3$ and H$_2$O mixing ratios in different seasons at the urban and suburban sites in Beijing.


In spring, the average diurnal variations in the NH$_3$ mixing ratio were similar at the urban and suburban sites. The diurnal variations exhibited a single-peak pattern with high values in the daytime and low values at night. The NH$_3$ mixing ratio began to increase in the morning, reached its maximum value

at 16:00, and then decreased gradually. The lowest mixing ratios at the urban and suburban sites occurred
at 03:00 and 09:00, respectively. The $NH_3$ mixing ratio began to increase earlier at the urban site than at
the suburban site. A plausible explanation to the earlier increase in the $NH_3$ emission at the urban site is
the traffic emission in the morning rush hours. In spring, the mixing ratio of $NH_3$ was higher at the
suburban site than that at the urban site, with an average difference of 4.1 ppb and a maximum difference
of 6.1 ppb. The average diurnal amplitude of the $NH_3$ mixing ratio at the suburban site was 5.3 ppb,
which was higher than that (2.6 ppb) at the urban site. At the urban site, the average diurnal variations in
the $NH_3$ and $H_2O$ mixing ratios exhibited nearly opposite trends. The $H_2O$ mixing ratio had high values
in the night and low values in the day. At the suburban site, the variation characteristics of $NH_3$ and $H_2O$
were very similar; however, the peak $NH_3$ concentration occurred 5 hours earlier than the peak $H_2O$
concentration. In spring, in contrast to the $NH_3$ mixing ratio, the $H_2O$ mixing ratio at the urban site was
1279 ppm higher than that at the suburban site.

The diurnal variation in the $NH_3$ mixing ratio at the suburban site in summer was similar to that in

spring. This phenomenon was also observed in the rural areas of Shanghai by Chang et al. (2019). The
diurnal variations of $NH_3$ at the suburban site were considerably affected by the temperature and the
contribution from volatile $NH_3$ sources. However, the diurnal summer variation of $NH_3$ at the urban site
was completely different from that at the suburban site. The summer level of $NH_3$ at the urban site was
obviously lower during the daytime and evening than that at the suburban site, increased gradually from
21:00 to levels higher its suburban counterpart, dropped after reaching its peak value at 7:00, and then
reached its lowest value at 14:00. The diurnal pattern (with a peak in early morning) has been observed
in other areas, such as rural (Ellis et al., 2011), urban (Gong et al., 2011), and steppe areas located far
away from human activity (Wentworth et al., 2016). Kuang et al. (2020) believed that such diurnal pattern
of $NH_3$ was caused by the evaporation of dew in the morning, which resulted in the release of $NH_3$
originally stored in the droplets. A lag was observed between the changes in the $NH_3$ and $H_2O$
concentrations in the early morning, which supported the hypothesis of Kuang et al (2020). In addition,
the increase in the $NH_3$ concentration in the morning might have been caused by the breakup of the
boundary layer formed at night. The downward mixing of air with a higher $NH_3$ concentration in the
residual layer led to a morning increase in the $NH_3$ concentration on the ground (Bash et al., 2010). In
summer, the $NH_3$ concentrations at the suburban site were significantly higher than those at the urban
site during the daytime and first half of the night. The average diurnal amplitude of the $NH_3$ concentration
was 7.5 ppb and 3.7 ppb at the urban and suburban sites, respectively. Similar to the situation in spring,
the $H_2O$ concentrations at the urban site were significantly higher than those at the suburban site in the
summer.

In autumn, the $NH_3$ concentration at the suburban site was relatively stable and remained nearly all

the time lower than that at the urban site, which showed low values during the day and high values during
the night, with a peak at midnight and a minimum (about 2.0 ppb lower than the peak) at 17:00. The $H_2O$
concentration was marginally lower (250 ppm) at the urban site than at the suburban site. The diurnal
profiles of $H_2O$ at both sites resemble that of $NH_3$ at the urban site, but the lowest values of $H_2O$ occurred
earlier than the lowest value of $NH_3$ at the urban site.

The diurnal patterns of $NH_3$ and $H_2O$ in winter were similar to those in autumn though the mixing

ratios of $NH_3$ and $H_2O$ were lower than their autumn counterparts. There were two slight differences: (1)
the mixing ratios of $NH_3$ at both sites exhibited lower fluctuations than those in autumn and (2) the
mixing ratio of $NH_3$ at the urban site reached its minimum in winter earlier than that in autumn.

The above results indicate that although the two sites were under the influence of similar weather

systems, the diurnal variations in the $NH_3$ mixing ratios at the two sites were different in different seasons.
This finding suggests that different $NH_3$ sources and possibly sinks had different contributions to the $NH_3$
concentrations at the urban and suburban sites. Additional studies should be conducted to better
understand the behaviors of atmospheric $NH_3$ and its influencing factors.
*3.4. Effect of meteorological factors on the $NH_3$ levels*
Table 3 presents the annual and seasonal correlation coefficients between the daily means of $NH_3$
mixing ratios and those of the temperature, relative humidity, and wind speed at the two sites. Annually,
the $NH_3$ mixing ratios at both sites were positively correlated with temperature and relative humidity and
negatively correlated with wind speed, and the correlations are all highly significant. However, the
correlations deteriorated somewhat in warm seasons. In summer and autumn, no significant correlations
were noted between ambient $NH_3$ and temperature at the two sites. The correlation between $NH_3$ and
wind speed in summer was much weaker than in the other seasons. The relative humidity was stronger
correlated with the $NH_3$ concentration at the two sites than temperature, which can be perceived in Fig 2.
Also, the correlation between $NH_3$ and relative humidity did not vary much from season to season. This
implies a possibility that relative humidity exerts a certain influence on the variation of the $NH_3$ level in
the surface layer.

Table 3. Correlations between the daily mean values of $NH_3$ and meteorological elements (Spearman's
rank correlation coefficient)

| Site | Time Period | Temperature | Relative humidity | Wind speed |
|------|-------------|-------------|-------------------|------------|
|      | Annual      | 0.680**     | 0.706**           | -0.370**   |
|      | Spring      | 0.450**     | 0.645**           | -0.540**   |
| Urban | Summer     | 0.043       | 0.488**           | -0.106**   |
|      | Autumn      | 0.101       | 0.759**           | -0.413**   |
|      | Winter      | 0.596**     | 0.690**           | -0.449**   |

| | | | | |
|---|---|---|---|---|
| | Annual | 0.745** | 0.730** | -0.325** |
| | Spring | 0.256* | 0.518** | -0.391** |
| Suburban | Summer | 0.126 | 0.576** | -0.061** |
| | Autumn | 0.135 | 0.792** | -0.618** |
| | Winter | 0.676** | 0.663** | -0.545** |

*: at the 0.05 significant level; **: at the 0.01 significant level.

Fig. 5 displays the seasonal mean diurnal variations in the $NH_3$ mixing ratio, temperature, and
relative humidity in different seasons at the urban and suburban sites, with their correlation coefficients
shown in Fig. S5. At the urban site, the seasonal-hourly means of the $NH_3$ mixing ratio were positively
(negatively) correlated with those of temperature (relative humidity) in spring, but the correlations were
reversed in the other seasons. At the suburban site, the seasonal-hourly means of the $NH_3$ mixing ratio
were positively (negatively) correlated with those of temperature (relative humidity) in the spring and
summer, but less correlated in autumn and winter. Similar correlation behaviors (diurnal variations) were
found at both sites in spring, but in other seasons the correlations (diurnal variations) at the urban site
behaved differently from those at the suburban site. The inconsistent behaviors in summer, autumn and
winter caused urban-suburban differences in the annual-diurnal patterns of $NH_3$, temperature and relative
humidity as well as the $NH_3$-temperature (relative humidity) correlations, as can be seen in Fig. S6.
Figure 6 displays the contour maps of the $NH_3$ mixing ratio, temperature, and relative humidity in
different seasons at the urban and suburban sites. The annual contour maps are shown in Fig. S7. As
shown in these contour maps, the $NH_3$ mixing ratios at both sites increased with relative humidity at
same temperature and increased with temperature at same relative humidity. Although there are some
scatterings in the contour maps, high $NH_3$ levels are generally associated with high temperature and
humidity. In winter, when air temperature was low ($< 0$ °C), the $NH_3$ mixing ratios at both sites often had
low values except in high humidity (>60%). An increase in temperature caused higher $NH_3$ mixing ratios
at both sites; however, the NH₃ concentration at the suburban site was more significantly correlated with
temperature than that at the urban site (Table 3), suggesting that volatile NH₃ sources might have a higher
contribution to the NH₃ concentration in suburban than in urban area. A higher amount of NH₃ removal
through chemical transformation is expected during the day at the urban site than at the suburban site
because the urban area had higher relative humidity and amounts of particulate matters, and higher
emissions of acid gases (particularly $NO_x$) than the suburban area. In 2018, the concentrations of $PM_{2.5}$,
$SO_2$ and $NO_2$ were 50 μg/m³, 5 μg/m³, 43μg/m³ in Haidian, and 46 μg/m³, 6 μg/m³, 35 μg/m³ in
Changping, respectively, as reported by Beijing Ecology and Environment Statement.

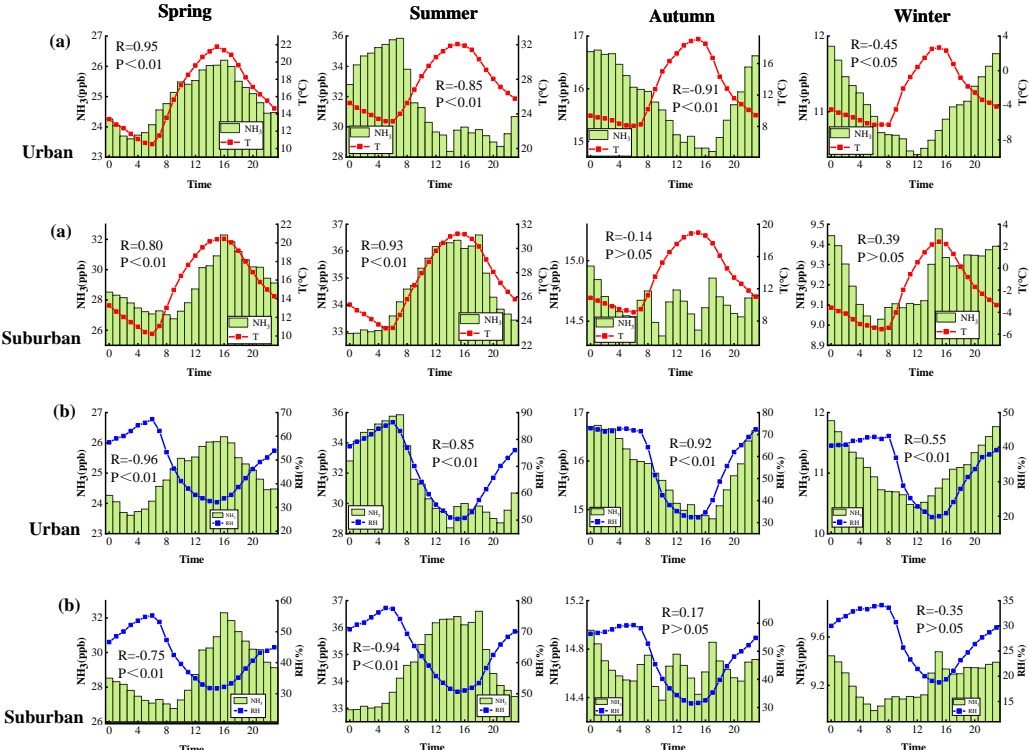

**Fig. 5.** Diurnal variations in and correlation coefficients between the NH₃ mixing ratios and temperature (a), relative humidity (b) in
different seasons at the urban and suburban sites in Beijing.

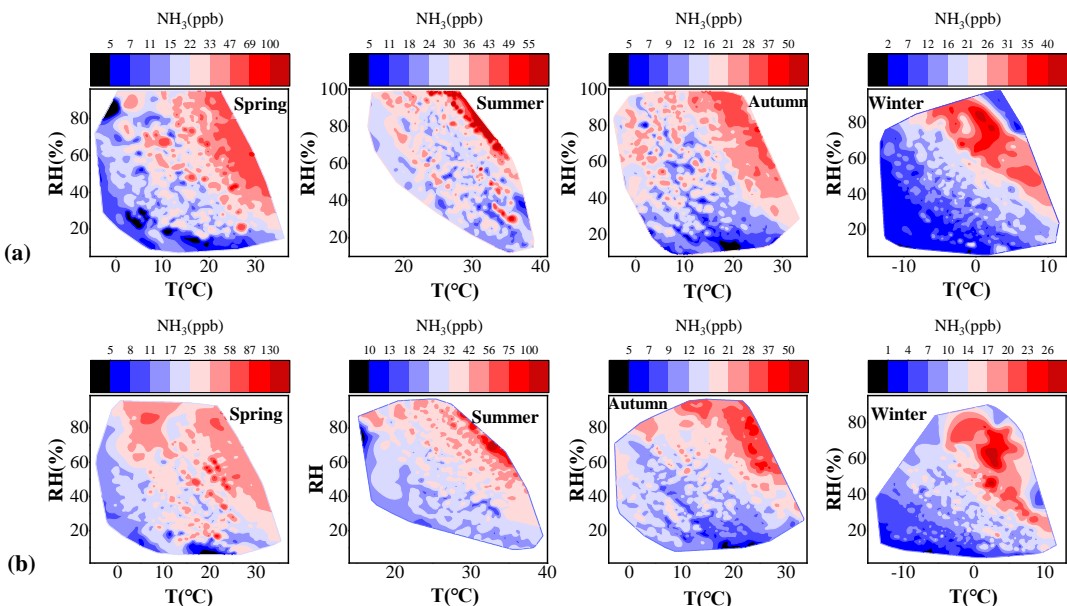

**Fig. 6.** Contour maps of the $NH_3$ mixing ratio, temperature, and relative humidity in different seasons at the urban and suburban sites in Beijing (a: Urban, b: Suburban).

To explore the influence of wind on the $NH_3$ mixing ratios, rose charts were drawn for the hourly mean concentration of $NH_3$, wind direction frequency, and wind speed during the observation period (Fig. 7). The large-scale wind circulation in the North China Plain is often influenced by the mountain-plain topography; therefore, the dominant winds in this region are southerly (from noon to midnight) and northerly (from midnight to noon) (Lin et al., 2009; Lin et al., 2011). As displayed in Fig. 7, some differences existed in the distributions of the surface wind between the urban and suburban sites. The prevailing surface winds were northeasterly and southwesterly at the urban site and northwesterly and easterly at the suburban site. At the urban site, the $NH_3$ mixing ratios were relatively high when the winds originated from the southern sectors and relatively low when the winds originated from the northwest sectors. Therefore, under southwest wind, air masses from the south of Beijing carry not only air pollutants but also higher levels of $NH_3$ to the urban site. Meng et al. (2017) examined the effect of long-

range air transport on the urban NH$_3$ levels in Beijing during the summer through trajectory analysis.

They concluded that the air mass from the southeast has a cumulative effect on the NH$_3$ concentration.

Although the dominant wind direction at the suburban site was different from that at the urban site, the

NH$_3$ mixing ratios were also relatively high in the south sectors. Thus, winds from the southeast, south,

and southwest can elevate levels of atmospheric NH$_3$ at both the urban and suburban sites. The NH$_3$

mixing ratios were relatively low when air masses originated from the northwest sector at urban site and

from the west sector at the suburban site. The west and northwest winds were stronger and promoted the

dilution and diffusion of NH$_3$ emitted into the boundary layer.

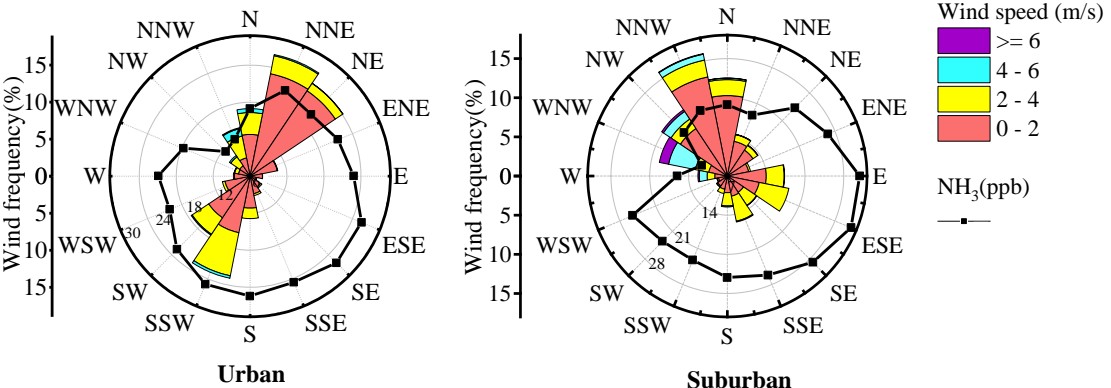

**Fig. 7.** Rose maps of the NH$_3$ mixing ratios, wind frequency, and wind speed in different wind direction sectors.

As a water-soluble gas, NH$_3$ can be impacted by precipitation. Heavy rainfall occurred on August

18, 2018 (Fig. 8). Before the rainfall, the NH$_3$ concentration at the urban site was higher than the average

level in August. After the rainfall, the NH$_3$ concentration decreased rapidly, and it was significantly lower

than the mean value in August. However, the diurnal pattern of NH$_3$ on that day did not differ

considerably from the average diurnal pattern in August. On the same day, the NH$_3$ mixing ratio at the

suburban site remained at a low level during the rainfall period, which was considerably lower than the

August mean NH$_3$ concentration during the same time of day. However, the NH$_3$ mixing ratio increased

rapidly after the precipitation and reached the mean level at 17:00. The rainfall might have an obvious
clearing effect on $NH_3$ but more case studies are needed to reach a robust conclusion.

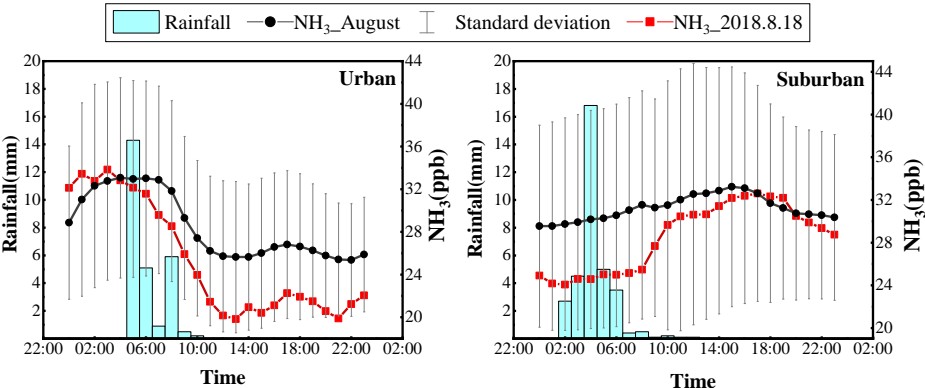

**Fig. 8.** Diurnal variations in the rainfall and $NH_3$ concentration on August 18, 2018.

**4. Conclusions**

In this study, the atmospheric $NH_3$ concentrations at an urban site and a suburban site in Beijing

were continuously and simultaneously observed from January 2018 to January 2019. The mean $NH_3$
mixing ratios were $21 \pm 14$ ppb and $22 \pm 15$ ppb at the urban and suburban sites, respectively. These $NH_3$
levels are among the highest mean values found in China and much higher than those reported for some
developed countries in America, Europe and Asia. In the summer and spring, the $NH_3$ mixing ratios at
the suburban site were slightly higher than those at the urban site. In the autumn and winter, however,
the situation was reversed. The highest $NH_3$ mixing ratios at the urban and suburban sites were all found
in July. The lowest $NH_3$ mixing ratio occurred in February at the urban site and in January at the suburban
site. A comparison with data from literature shows that the mean concentration of $NH_3$ in Beijing did not
change considerably in the decade before 2019.

The hourly mean $NH_3$ mixing ratios at the urban site were highly correlated ($R = 0.849$, $P < 0.01$)

with those at the suburban site. However, the mean diurnal variations in the $NH_3$ mixing ratios at the
urban and suburban sites were different. At the urban site, lower NH$_3$ mixing ratios were observed in the
daytime and higher ones at night. The opposite trend was observed at the suburban site. Although both
sites were under the influence of similar weather systems, the seasonal-diurnal variations in the NH$_3$
mixing ratio were different at the urban and suburban sites, suggesting that NH$_3$ sources had different
relative contributions to the NH$_3$ levels at the urban and suburban sites.
The relationship of meteorological factors with the NH$_3$ mixing ratio was complex. Overall, the NH$_3$
mixing ratios increased with relative humidity and temperate at both sites. Relative humidity was stronger
correlated with the NH$_3$ mixing ratio at both sites. The situation in different seasons varied and was site-
dependent, which warrants further studies. A high wind speed (mainly under northwesterly) suppressed
the levels of NH$_3$ at both sites. The NH$_3$ mixing ratios were higher under southerly wind conditions.
Rainfall had a certain scavenging effect on NH$_3$ but had little effect on the diurnal variations in the NH$_3$
concentration.

*Data availability.* The data of stationary measurements are available upon request to the contact author
Weili Lin (linwl@muc.edu.cn).

*Author contributions.* ZL and WL developed the idea for this paper, formulated the research goals, and
carried out the measurement at urban site. WP and ZM carried out the NH$_3$ field observations at the
suburban site.

**Competing interests.** The authors declare that they have no conflict of interest.

**Acknowledgments.** This study was funded by the National Natural Science Foundation of China
(Grant No. 91744206) and the Beijing Municipal Science and Technology (Z181100005418016).

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

Contributions of Agricultural and Nonagricultural Emissions to Atmospheric Ammonia in a

Chinese Megacity, Environmental Science and Technology, 53(4), 1822–1833,

doi:10.1021/acs.est.8b05984, 2019.

Charlson, R.J., LANGNER, J., Rodhe, H., Leovy, C.B., Warren, S.G.: Perturbation of the northern

hemisphere radiative balance by backscattering from anthropogenic sulfate aerosols, Tellus B:

Chemical and Physical Meteorology, 43(4),12, doi:10.1034/j.1600-0889.1991.t01-1-00013.x,

1991.

Edgerton, E. S., Saylor, R. D., Hartsell, B. E., Jansen, J. J. and Alan Hansen, D.: Ammonia and

ammonium measurements from the southeastern United States, Atmospheric Environment, 41(16),

3339-3351, doi:10.1016/j.atmosenv.2006.12.034, 2007.

Ellis, R. A., Murphy, J. G., Markovic, M. Z., Vandenboer, T. C., Makar, P. A., Brook, J. and Mihele, C.:

The influence of gas-particle partitioning and surface-atmosphere exchange on ammonia during

BAQS-Met, Atmospheric Chemistry and Physics, 11(1), 133-145, doi:10.5194/acp-11-133-2011,

2011.

Erisman, J. W., Bleeker, A., Galloway, J. and Sutton, M. S.: Reduced nitrogen in ecology and the

environment, Environmental Pollution, 150(1), 140-149, doi:10.1016/j.envpol.2007.06.033, 2007.

Fu, X., Wang, S., Xing, J., Zhang, X., Wang, T. and Hao, J.: Increasing Ammonia Concentrations

Reduce the Effectiveness of Particle Pollution Control Achieved via $SO_2$ and $NO_X$ Emissions

Reduction in East China, Environmental Science and Technology Letters, 4(6), 221–227,

doi:10.1021/acs.estlett.7b00143, 2017.

Ge, B., Xu, X., Ma, Z., Pan, X., Wang, Z., Lin, W., Ouyang, B., Xu, D., Lee, J., Zheng, M., Ji, D., Sun,

Y., Dong, H., Squires, F.A., Fu, F., Wang, Z.: Role of ammonia on the feedback between AWC and

inorganic aerosol formation during heavy pollution in the North China Plain, Earth and Space

Science, 6, 1675-1693, https://doi.org/10.1029/2019EA000799, 2019.

Gong, L., Lewicki, R., Griffin, R. J., Flynn, J. H., Lefer, B. L. and Tittel, F. K.: Atmospheric ammonia

measurements in Houston, TX using an external-cavity quantum cascade laser-based sensor,

Atmospheric Chemistry and Physics, 11(18), 9721–9733, doi:10.5194/acp-11-9721-2011, 2011.

He, Y., Pan, Y., Zhang, G., Ji, D., Tian, S., Xu, X., Zhang, R. and Wang, Y.: Tracking ammonia morning

peak, sources and transport with 1 Hz measurements at a rural site in North China Plain,

Atmospheric Environment, 235, https://doi.org/10.1016/j.atmosenv.2020.117630, 2020.

Hu, Q., Zhang, L., Evans, G. J. and Yao, X.: Variability of atmospheric ammonia related to potential

emission sources in downtown Toronto, Canada, Atmospheric Environment, 99,

doi:10.1016/j.atmosenv.2014.10.006, 2014.
Huang, X., Song, Y., Li, M., Li, J., Huo, Q., Cai, X., Zhu, T., Hu, M. and Zhang, H.: A high-resolution
ammonia emission inventory in China, Global Biogeochemical Cycles, 26(1),
doi:10.1029/2011GB004161, 2012.
Ju, X. T., Xing, G. X., Chen, X. P., Zhang, S. L., Zhang, L. J., Liu, X. J., Cui, Z. L., Yin, B., Christie, P.,
Zhu, Z. L. and Zhang, F. S.: Reducing environmental risk by improving N management in
intensive Chinese agricultural systems, Proceedings of the National Academy of Sciences of the
United States of America, 106(9), 3041-3046, doi:10.1073/pnas.0813417106, 2009.
Krotkov, N.A., McLinden, C.A., Li, C., Lamsal, L.N., Celarier, E.A., Marchenko, S. v., Swartz, W.H.,
Bucsela, E.J., Joiner, J., Duncan, B.N., Boersma, K.F., Veefkind, J.P., Levelt, P.F., Fioletov, V.E.,
Dickerson, R.R., He, H., Lu, Z., Streets, D.G.: Aura OMI observations of regional $SO_2$ and $NO_2$
pollution changes from 2005 to 2015. Atmospheric Chemistry and Physics 16(7), 4605–4629,
doi:10.5194/acp-16-4605-2016, 2016.
Kuang, Y., Xu, W., Lin, W., Meng, Z., Zhao, H., Ren, S., Zhang, G., Liang, L. and Xu, X.: Explosive
morning growth phenomena of NH3 on the North China Plain: Causes and potential impacts on
aerosol formation, Environmental Pollution, 257, 113621, doi:10.1016/j.envpol.2019.113621,

2020.

Liao, X., Zhang, X., Wang, Y., Liu, W., Du, J. and Zhao, L.: Comparative Analysis on Meteorological
Condition for Persistent Haze Cases in Summer and Winter in Beijing, Environmental Science,
35(06), 2031–2044, doi:10.13227/j.hjkx.2014.06.001, 2014.
Lin, W., Xu, X., Ge, B., Liu, X.: Gaseous pollutants in Beijing urban area during the heating period
2007-2008: variability, sources, meteorological and chemical impacts, Atmos. Chem. Phys., 11,

8157-8170, 2011.

Lin, W., Xu, X., Ge, B., Zhang, X.: Characteristics of gaseous pollutants at Gucheng, a rural site

southwest of Beijing, J. Geophys. Res., 114, D00G14, doi:10.1029/2008JD010339, 2009.

Meng, Z. Y., Lin, W. L., Jiang, X. M., Yan, P., Wang, Y., Zhang, Y. M., Jia, X. F. and Yu, X. L.:

Characteristics of atmospheric ammonia over Beijing, China, Atmospheric Chemistry and Physics,

11(12), 6139–6151, doi:10.5194/acp-11-6139-2011, 2011.

Meng, Z. Y., Xu, X. bin, Wang, T., Zhang, X. Y., Yu, X. L., Wang, S. F., Lin, W. L., Chen, Y. Z., Jiang,

Y. A. and An, X. Q.: Ambient sulfur dioxide, nitrogen dioxide, and ammonia at ten background

and rural sites in China during 2007–2008, Atmospheric Environment, 44(21–22), 2625-2631,

doi:10.1016/j.atmosenv.2010.04.008, 2010.

Meng, Z., Lin, W., Zhang, R., Han, Z. and Jia, X.: Summertime ambient ammonia and its effects on

ammonium aerosol in urban Beijing, China, Science of the Total Environment, 579, 1521–1530,

doi:10.1016/j.scitotenv.2016.11.159, 2017.

Meng, Z., Wu, L., Xu, X., Xu, W., Zhang, R., Jia, X., Liang, L., Miao, Y., Cheng, H., Xie, Y., He, J. and

Zhong, J.: Changes in ammonia and its effects on $PM_{2.5}$ chemical property in three winter seasons

in Beijing, China, Science of The Total Environment, 749, 142208,

doi:10.1016/j.scitotenv.2020.142208, 2020.

Nowak, J. B., Huey, L. G., Russell, A. G., Tian, D., Neuman, J. A., Orsini, D., Sjostedt, S. J., Sullivan,

500          A. P., Tanner, D. J., Weber, R. J., Nenes, A., Edgerton, E. and Fehsenfeld, F. C.: Analysis of urban

gas phase ammonia measurements from the 2002 Atlanta Aerosol Nucleation and Real-Time

Characterization Experiment (ANARChE), Journal of Geophysical Research Atmospheres,

111(17), doi:10.1029/2006JD007113, 2006.

Osada, K., Saito, S., Tsurumaru, H. and Hoshi, J.: Vehicular exhaust contributions to high $NH_3$ and

$PM_{2.5}$ concentrations during winter in Tokyo, Japan, Atmospheric Environment, 206, 218–224,

doi:10.1016/j.atmosenv.2019.03.008, 2019.

Pan, Y., Tian, S., Zhao, Y., Zhang, L., Zhu, X., Gao, J., Huang, W., Zhou, Y., Song, Y., Zhang, Q. and

Wang, Y.: Identifying Ammonia Hotspots in China Using a National Observation Network,

Environmental Science and Technology, 52(7), 3926–3934, doi:10.1021/acs.est.7b05235, 2018.

Pearson, J. and Stewart, G.R.: The deposition of atmospheric ammonia and its effects on plants, New

Phytologist, 125(2), 283–305, doi:10.1111/j.1469-8137.1993.tb03882.x, 1993.

Phan, N.-T., Kim, K.-H., Shon, Z.-H., Jeon, E.-C., Jung, K. and Kim, N.-J.: Analysis of ammonia

variation in the urban atmosphere, Atmospheric Environment, 65, 177–185,

https://doi.org/10.1016/j.atmosenv.2012.10.049, 2013.

Pinder, R. W., Gilliland, A. B. and Dennis, R. L.: Environmental impact of atmospheric $NH_3$ emissions

under present and future conditions in the eastern United States, Geophysical Research Letters,

35(12), 89-90, doi:10.1029/2008GL033732, 2008.

Pu, W., Ma, Z., Collett, J. L., Guo, H., Lin, W., Cheng, Y., Quan, W., Li, Y., Dong, F. and He, D.:

Regional transport and urban emissions are important ammonia contributors in Beijing, China,

Environmental Pollution, 265, doi:10.1016/j.envpol.2020.115062, 2020.

Reay, D. S., Dentener, F., Smith, P., Grace, J. and Feely, R. A.: Global nitrogen deposition and carbon

sinks, Nature Geoscience, 1(7), 430-437, doi:10.1038/ngeo230, 2008.

Saraswati, George, M. P., Sharma, S. K., Mandal, T. K. and Kotnala, R. K.: Simultaneous

Measurements of Ambient NH 3 and Its Relationship with Other Trace Gases, $PM_{2.5}$ and

Meteorological Parameters over Delhi, India, Mapan - Journal of Metrology Society of India,

34(1), 55–69, doi:10.1007/s12647-018-0286-0, 2019.

Singh, S. and Kulshrestha, U. C.: Rural versus urban gaseous inorganic reactive nitrogen in the Indo-Gangetic plains (IGP) of India, Environ. Res. Lett., 9(12), 125004, https://doi.org/10.1088/1748-9326/9/12/125004, 2014.

UN Environment 2019. A Review of 20 Years' Air Pollution Control in Beijing. United Nations Environment Programme, Nairobi, Kenya. https://www.unenvironment.org/resources/report/review-20-years-air-pollution-control-beijing.

Van Breemen, N., Mulder, J. and Driscoll, C. T.: Acidification and alkalinization of soils, Plant and Soil, 75(3), https://doi.org/10.1007/BF02369968, 1983.

Vogt, E., Held, A. and Klemm, O.: Sources and concentrations of gaseous and particulate reduced nitrogen in the city of Münster (Germany), Atmospheric Environment, 39(38), 7393–7402, https://doi.org/10.1016/j.atmosenv.2005.09.012, 2005.

Wang, K., Fan, S., Guo, J. and Sun, G.: Characteristics of ammonia emission from motor vehicle exhaust in Beijing, Environmental Engineering, 36(03), 98–101, doi:10.13205/j.hjgc.201803020, 2019.

Warner, J. X., Dickerson, R. R., Wei, Z., Strow, L. L., Wang, Y. and Liang, Q.: Increased atmospheric ammonia over the world's major agricultural areas detected from space, Geophysical Research Letters, 44(6), 2875–2884, doi:10.1002/2016GL072305, 2017.

Wei, S., Dai, Y., Liu, B., Zhu, A., Duan, Q., Wu, L., Ji, D., Ye, A., Yuan, H., Zhang, Q., Chen, D., Chen, M., Chu, J., Dou, Y., Guo, J., Li, H., Li, J., Liang, L., Liang, X., Liu, H., Liu, S., Miao, C. and Zhang, Y.: A China data set of soil properties for land surface modeling, Journal of Advances in Modeling Earth Systems, 5(2), 212–224, doi:10.1002/jame.20026, 2013.

Wentworth, G. R., Murphy, J. G., Benedict, K. B., Bangs, E. J. and Collett, J. L.: The role of dew as a
night-time reservoir and morning source for atmospheric ammonia, Atmospheric Chemistry and
Physics, 16(11), 7435–7449, doi:10.5194/acp-16-7435-2016, 2016.
Wu, Z., Hu, M., Shao, K. and Slanina, J.: Acidic gases, NH3 and secondary inorganic ions in $PM_{10}$
during summertime in Beijing, China and their relation to air mass history, Chemosphere, 76(8),
doi:10.1016/j.chemosphere.2009.04.066, 2009.
Yao, X. and Zhang, L.: Trends in atmospheric ammonia at urban, rural, and remote sites across North
America, Atmos. Chem. Phys., 16(17), 11465–11475, https://doi.org/10.5194/acp-16-11465-2016,

2016.

Zhang, B.: Atmospheric Distribution and Variation of $NH_3$ in Beijing, Environmental Science and
Management 41(01), 119–122, 2016.
Zhang, S., Wag, A., Zhang, Z., Wang, J., Han, Y., Su, R. and Qu, Y.: On creating an anthropogenic
ammonia emission inventory in capital Beijing, Journal of Safety and Environment, 16(02), 242–
245, doi:10.13637/j.issn.1009–6094.2016.02.047, 2016.
Zhang, X., Wu, Y., Liu, X., Reis, S., Jin, J., Dragosits, U., van Damme, M., Clarisse, L., Whitburn, S.,
Coheur, P. F. and Gu, B.: Ammonia emissions may be substantially underestimated in China,
Environmental Science and Technology, 51(21), 12089–12096, doi:10.1021/acs.est.7b02171,

2017.

Zhang, Y., Tang, A., Wang, D., Wang, Q., Benedict, K., Zhang, L., Liu, D., Li, Y., Collett Jr., J. L., Sun,
Y. and Liu, X.: The vertical variability of ammonia in urban Beijing, China, Atmospheric
Chemistry and Physics, 18(22), 16385–16398, doi:10.5194/acp-18-16385-2018, 2018.
Zhao, X., Xie, Y. X., Xiong, Z. Q., Yan, X. Y., Xing, G. X. and Zhu, Z. L.: Nitrogen fate and
environmental consequence in paddy soil under rice-wheat rotation in the Taihu lake region,
China, Plant and Soil, 319(1), 225-234, doi:10.1007/s11104-008-9865-0, 2009.
Zhou, C., Zhou, H., Holsen, T. M., Hopke, P. K., Edgerton, E. S. and Schwab, J. J.: Ambient Ammonia
Concentrations Across New York State, Journal of Geophysical Research: Atmospheres, 124(14),
8287–8302, doi:10.1029/2019JD030380, 2019.