# Peer review of "Measurement report: Exploring the NH3 behaviors at urban and suburban Beijing"

_Atmospheric Chemistry and Physics, 2020_

## Referee Comment (RC2)

Comments on "Measurement report: Exploring the NH3 behaviours at urban and suburban Beijing: Comparison and implications"

General comments:

This manuscript reports a yearlong parallel measurements of NH3 concentration at urban of suburban sites of Beijing. Statistical analysis, seasonal variation, diel profile and the relationship with temperature, RH, H2O, wind speed and wind direct are present. This could be useful documentary for scientists who are interesting in aerosol chemistry, aerosol pH and pollutant management etc. I thus recommend publication of this manuscript on Atmospheric Chemistry and Physics.

Specific comments:

1. The author might be able to take advantage of the parallel measurements and summarize the difference in pattern of $NH_3$ pollution and source. However, there seems to be a lack of effort on the in-depth analysis.

2. The dew and NH3 releasing point is interesting. I wonder if the dew point calculation could be useful to further explore and validate this point.

3. The author claimed a rain clearing effect for NH3. But in Figure 8 suburban case, there was an increasing during the rain and after the rain. Therefore, there is a inconsistence between the data and the explanation.

---

## Short Comment (SC1) · 10 Nov 2020

The paper present the continuous measurement of mixing ratio of ambient NH3 at an urban site and a suburban site of Beijing, China from January 13, 2018, to January 13, 2019 alongwith its interaction with meteorology (temperature, relative humidity, wind speed and wind direction). It is also reported that the same temperature (relative humidity) at the urban and suburban sites, the NH3 mixing ratios increased with the relative humidity (temperature), whereas a high wind speed promoted a reduction in the NH3 mixing ratio. There are a number of studies on ambient NH3 and its interaction with other trace gases (NO, NO2, CO, SO2, and VOCs etc,) and meteorology

have been carried out over the China, other Asian region and abound the globe. Some of the papers published in Asian regions are highlighted in the Introduction section, however, some of recent publications/studies are mission which is need to be updated. The present paper is not adding much more new scientific information, except location specific information alongwith meteorology.

Some of the suggestions are

- The "mixing ratio" and "concentration" are two different term. Hence, use only mixing ratio of NH3 in entire text.

- "The NH3 mixing ratio increased with the relative humidity (temperature), whereas a high wind speed promoted a reduction in the NH3 mixing ratio" for this statement provide the more appropriate references/support. Apart from the influence of meteorological condition, the source strength of the observational sites are also important, which is not explained in the text.

- In the present study, authors have reported only mixing ratio of NH3 and meteorology. The observations of other precursors gases (NO, NO2, CO, and SO2 etc,) of NH3 is missing in the present study. These precursors gases (NO, NO2, CO, and SO2 etc,) are more important to estimate the mixing ratio of NH3 for a particular loaction.

- Table 1 need to be updated with more study of various megacities of the world. a lot of Informations are available for China and Indian region. Authors may be go through the following review paper for the comparisons of Table 1

"Sharma, S.K., Kotnala, G., and Mandal, T.K., (2020). Spatial variability and sources of atmospheric ammonia in India: a review. Aerosol Science & Engineering, Vol. 4(1), 1-8"

- Some of the explanations are mentioned in terms of "concentration ($\mu$g/m3)". Line No. 260: "In 2018, the concentrations of PM2.5, SO2 and NO2 were 50$\mu$g/m3, 5$\mu$g/m3, 43$\mu$g/m3 in Haidian, and 46$\mu$g/m3, 6$\mu$g/m3, 35$\mu$g/m3". Use either mixing ratios or

concentration and convert the values accordingly to make them more informative not confusing term.

---

## Referee Comment (RC1) · Anonymous Referee #1 · 29 Nov 2020

Overview:

The paper is well written and presents a very important dataset which adds to the rather sparse number of long term ambient ammonia datasets. The information provided gives a detailed insight to the ammonia variation at the two sites and the influence of meteorological conditions. However the context of the measurement and the emissions environment driving the concentrations would make for a more powerful paper. In addition there needs to be greater detail in the analytical methodology and presentation of quantitative calibration, relevant uncertainties and the analytical method for interpretation of the measurement which hopefully the authors can provide which

would strength the paper.

Detailed comments:

Section 2.1: It would be very useful if there is an emissions or land use map for each of the sites, for the surrounding area in addition to the large scale map.

Section 2.2 It is clear that the authors have taken care to calibrate the ammonia monitor. However unfortunately no calibration data is presented which it should be. Firstly it is stated that "obtained concentration was normalized with respect to a reference concentration". The normalisation factor should be reported. The detection limit of the instrument is noted as 0.2 however given that the authors discuss the issue of ammonia stickiness, the baseline would need to be checked for drift regularly in order to identify any baseline drift particularly as the internal surfaces of the instrument become coated. Was this done through out the deployments? Was there any evidence of baseline drift? It is quite important to show this data so that the reader can have confidence in the reported concentrations. Presenting the calibration data and blanks (ammonia-free air) would be very useful for the reader to have confidence in the accuracy of the data presented.

The set up of the instrument is described but key parameters including the flow rates are not reported. What was the sampling flow? Was an external pump used, and did the inlet lines sample directly from the air or pass through a manifold? It is noted that filters are used, but are the filters changed regularly or cleaned to remove particulate (which can revolatilise NH3 if warmed). Are the filters temperature stabilised?

Also it would be interesting to know how the calibration was done with a standard gas cylinder. The experience of the reviewer has been that the addition of a dry calibrated reference NH3 gas leads to a complete loss of the NH3 signal in some of these OA-ICOS types of instruments (as the instrument uses the water line for holding the NH3 signal). Did the authors observe this? Was a humidified calibration done. The details of this would be useful information for the readers.

In order to understand the response of the instrument, it would be good to have a figure with some of the measured data before averaging is done, particularly during the case study of the precipitation event perhaps or during Spring when the ammonia concentrations are highest. What was the estimated response time of the instrument? It is true that the response time is faster than when going from low to high concentrations, but it would be useful for the authors to characterise that for this setup – it is the response of the sample lines as well as the instrument. From the long term dataset a precision and accuracy and LOD should be presented.

Results and Discussion: Figure 2: Please move baseline to zero as it is hard to interpret the NH3 concentrations at the low end. I would prefer the NH3 to be on it s own graphs so that the reader can easily look at the NH3 data which is the primary focus of the paper.

P6-7: Soil: There is some discussion around soil emissions of ammonia. I think it would be useful for the authors to make clear that soil itself does not emit ammonia per se, ammonia emissions from soils or vegetation are due to either fertiliser applied to the soil or ammonia deposition and re-emission. It would also be good to frame the discussion on the acidic soils with the aqueous acid-base chemistry, of which there is quite a big literature.

Section 3.2 and Section 3.3 and Supplementary material: The discussion of the differences in the seasonal and diurnal variability of NH3 and H2O is really interesting and highlights the importance of understanding the boundary layer height and dilution/dispersion processes driving ambient ammonia concentrations – which are very high for an altitude of 70-100m! Though outside of the scope of this paper, a detailed modelling study of the datasets could be very insightful. However because of the complexities and rather small concentration changes I think that both of these sections need to be more conservative about the changes/trends/drivers of changes in concentrations. I would suggest through out that they authors should highlight uncertainties in the analysis.

In particular the use of linear correlations is difficult to justify. In the supplementary material the linear correlation between T and RH of the NH3 are plotted and the correlations summarised in Section 3.3. I am a bit concerned about this simple approach. There is much evidence that the ammonia equilibrium in the environment is non-linear – specifically it is exponential with equilibrium concentrations doubling for $\sim 5oC$ in thermodynamic equilibrium. So rather than start with linear correlations the exponential model should be tested first (as the best theoretical relationship which has basis in physical chemistry). Another concern was that the relationship was been studied over very small concentration ranges (<2 ppb in some cases.). Once the authors have assessed the precision and accuracy of their dataset, then error bars can be applied to these plots and then in some cases no line should be plotted, or a larger data group analysed. A non-linear relationship can be seen in some of the graphs. The opposite relationship is seen in a couple of plots, therefore it would be useful for the authors to look at those in a bit more detail. I would suggest a review of this section to make more clearly justifiable statistical analyses between ammonia concentration, relative humidity and temperature.

Minor point: Some language checking would be useful.

---

## Author Comment (AC1) · 7 Feb 2021

Response to Interactive comment on "Measurement report: Exploring the NH3 behaviours at urban and suburban Beijing: Comparison and implications" by Ziru Lan et al.

Ziru Lan1, Weili Lin1, Weiwei Pu2, and Ziqiang Ma2,3

1College of Life and Environmental Sciences, Minzu University of China, Beijing 100081 2Environmental Meteorological Forecast Center of Beijing-Tianjin-Hebei, Beijing, 100089, China 3Beijing Shangdianzi Regional Atmosphere Watch Station, Beijing,

101507, China

— We thank for the constructive comment and we revised our manuscript according to the suggestions.

The paper presents the continuous measurement of mixing ratio of ambient NH3 at an urban site and a suburban site of Beijing, China from January 13, 2018, to January 13, 2019 along with its interaction with meteorology (temperature, relative humidity, wind speed and wind direction). It is also reported that the same temperature (relative humidity) at the urban and suburban sites, the NH3 mixing ratios increased with the relative humidity (temperature), whereas a high wind speed promoted a reduction in the NH3 mixing ratio. There are a number of studies on ambient NH3 and its interaction with other trace gases (NO, NO2, CO, SO2, and VOCs etc,) and meteorology have been carried out over the China, other Asian region and around the globe. Some of the papers published in Asian regions are highlighted in the Introduction section, however, some of recent publications/studies are mission which is need to be updated. The present paper is not adding much more new scientific information, except location specific information along with meteorology. Response: We add more information about NH3 in the revised introduction. Some of the suggestions are - The "mixing ratio" and "concentration" are two different term. Hence, use only mixing ratio of NH3 in entire text.

Response: Agreed. But, in context, sometimes the word of concentration is needed.

- The NH3 mixing ratio increased with the relative humidity (temperature), whereas a high wind speed promoted a reduction in the NH3 mixing ratio" for this statement provide the more appropriate references/support. Apart from the influence of meteorological condition, the source strength of the observational sites are also important, which is not explained in the text.

Response: The statement is based on the analysis between the NH3 mixing ratio and meteorological factors. There are no direct local sources influencing the observational

sites. The source strengths in Beijing can found in those researches on emission inventory. We add some information in the revised version.

- In the present study, authors have reported only mixing ratio of NH3 and meteorology. The observations of other precursors gases (NO, NO2, CO, and SO2 etc,) of NH3 is missing in the present study. These precursors gases (NO, NO2, CO, and SO2 etc,) are more important to estimate the mixing ratio of NH3 for a particular loaction.

Response: In this paper, we report one-year measurements of NH3 simultaneously at the urban and suburban areas in Beijing with high temporal resolution analyzers. There are much more data and analysis for the routine measurement of NO, NO2, CO, and SO2 than NH3 in Beijing. Since NO, NO2, CO, and SO2 are not precursors gases of NH3, we think the comment may concern the NH3 variations related with other gases. We would carry out the research later but not in this paper.

- Table 1 need to be updated with more study of various megacities of the world. A lot of informations are available for China and Indian region. Authors may be go through the following review paper for the comparisons of Table 1 "Sharma, S.K., Kotnala, G., and Mandal, T.K., (2020). Spatial variability and sources of atmospheric ammonia in India: a review. Aerosol Science & Engineering, Vol. 4(1),1-8".

Response: Thank you for your suggestion, we add some data in mega cities of the world.

- Some of the explanations are mentioned in terms of "concentration ($\mu$g/m3)". Line No. 260: "In 2018, the concentrations of PM2.5, SO2 and NO2 were 50$\mu$g/m3, 5$\mu$g/m3, 43$\mu$g/m3 in Haidian, and 46$\mu$g/m3, 6$\mu$g/m3, 35$\mu$g/m3". Use either mixing ratios or concentration and convert the values accordingly to make them more informative not confusing term.

Response: Sorry. Not all pollutants can be described in term of ppb, for example of PM. There are no necessary to convert concentration ($\mu$g/m3) to mixing ratios because

we just compare the different pollution levels here.
* * *

---

## Author Comment (AC3) · 7 Feb 2021

—We thank both referees for their very constructive comments and suggestions. We revised our manuscript according to their comments and suggestions.

Response to comments by referee 2

Anonymous Referee #2

General comments: This manuscript reports a year long parallel measurements of NH3 concentration at urban of suburban sites of Beijing. Statistical analysis, seasonal variation, diel profile and the relationship with temperature, RH, H2O, wind speed and wind

direct are present. This could be useful documentary for scientists who are interesting in aerosol chemistry, aerosol pH and pollutant management etc. I thus recommend publication of this manuscript on Atmospheric Chemistry and Physics. Specific comments: 1. The author might be able to take advantage of the parallel measurements and summarize the difference in pattern of NH3 pollution and source. However, there seems to be a lack of effort on the in-depth analysis.

Response: Thanks for the suggestions. We present the differences in the seasonal and diurnal variability of NH3 and H2O at urban and suburban Beijing in this manuscript. The present spatial resolution of NH3 emission inventory is not sufficient to support source difference analysis. Many other data should be used together to understand the NH3 pollution and source, and this will be carried out in further studies.

2. The dew and NH3 releasing point is interesting. I wonder if the dew point calculation could be useful to further explore and validate this point.

Response: Here, we reference some possible explanations to the diurnal changes. We calculate the dew points in summer for urban site (Fig. R1). There has some relationship but not all the truth.

Figure R1. The average diurnal change in NH3, dew point, and H2O in summer at the urban Beijing.

3. The author claimed a rain clearing effect for NH3. But in Figure 8 suburban case, there was an increasing during the rain and after the rain. Therefore, there is a inconsistence between the data and the explanation.

Response: We notice the phenomenon that there was a little increasing during the rain, and on average it was lower than the mean NH3 concentration in the same time in August. We pointed that the diurnal variation of NH3 on the rain day did not differ considerably from the average diurnal variation in August. At the suburban site, the diurnal NH3 mixing ratio increased in the daytime, so there was an increasing after the

rain. We pointed that the rainfall MIGHT have a clearing effect on NH3 but needed
more cases to support.

[Figure]

[Figure]

Figure R1. The average diurnal change in NH₃, dew point, and H₂O in summer at the urban

Beijing.

**Fig. 1.**

---

## Author Response (AR1)

***Response to Interactive comments on*** "Measurement report: Exploring the NH₃ behaviours at urban and suburban Beijing: Comparison and implications" **by Lin** ***et al.***

Ziru Lan[1], Weili Lin[1], Weiwei Pu[2], and Ziqiang Ma[2,3]

[1]College of Life and Environmental Sciences, Minzu University of China, Beijing 100081

[2]Environmental Meteorological Forecast Center of Beijing-Tianjin-Hebei, Beijing, 100089, China

[3]Beijing Shangdianzi Regional Atmosphere Watch Station, Beijing, 101507, China

---We thank both referees for their very constructive comments and suggestions. We revised our manuscript according to their comments and suggestions.

**Response to comments by referee 1**

**Anonymous Referee #1**

Overview:

The paper is well written and presents a very important dataset which adds to the rather sparse number of long term ambient ammonia datasets. The information provided gives a detailed insight to the ammonia variation at the two sites and the influence of meteorological conditions. However the context of the measurement and the emissions environment driving the concentrations would make for a more powerful paper. In addition there needs to be greater detail in the analytical methodology and presentation of quantitative calibration, relevant uncertainties and the analytical method for interpretation of the measurement which hopefully the authors can provide which would strength the paper.

Detailed comments:

Section 2.1: It would be very useful if there is an emissions or land use map for each of the sites, for the surrounding area in addition to the large scale map.

Response: The resolution of current NH₃ emission data is insufficient to display the difference of the two sites. We add a land use map (Fig. R1) in the revised version.

[Figure]

Figure R1. Observation sites and the topography and land use in Beijing

Section 2.2 It is clear that the authors have taken care to calibrate they ammonia monitor. However unfortunately no calibration data is presented which it should be. Firstly it is stated that "obtained concentration was normalized with respect to a reference concentration". The normalisation factor should be reported. The detection limit of the instrument is noted as 0.2 however given that the authors discuss the issue of ammonia stickiness, the baseline would need to be checked for drift regularly in order to identify any baseline drift particularly as the internal surfaces of the instrument become coated. Was this done through out the deployments? Was there any evidence of baseline drift? It is quite important to show this data so that the reader can have confidence in the reported concentrations. Presenting the calibration data and blanks (ammonia free air) would be very useful for the reader to have confidence in the accuracy of the data presented.

Response:The word of normalization is not propriate here. We mean that we use a same calibration standard source to calibrate the instruments. The detection limit of the instrument is 0.2 and the maximum drift of 0.2 ppb/24hrs, which provided by the manufacturer, and we don't check it during the measurement. Due to ammonia stickiness, zero check needs more time to stabilize, we only did it during the multiple-points calibrations. Figure R2 is a typical multiple-points calibration of NH$_3$ analyzer with a cylinder standard gas.

[Figure]

Figure R2. A typical multiple-points calibration of NH$_3$ analyzer with a cylinder standard gas

The set up of the instrument is described but key parameters including the flow rates are not reported. What was the sampling flow? Was an external pump used, and did the inlet lines sample directly from the air or pass through a manifold? It is noted that filters are used, but are the filters changed regularly or cleaned to remove particulate (which can revolatilise NH$_3$ if warmed). Are the filters temperature stabilised?

Response:The inlet lines sampled the air through a manifold and the lengths of the line from manifolds to the analyzers are less than 2.0 m. The flow rates of the analyzers are over 0.4 L/min. The filters are changed every 2–3 weeks. The filters and the instruments are deployed in an air-conditioning room, so the filters temperature is relatively stabilized. Inlet heated is often suggested by people to reduce NH$_3$ "sticks" to surfaces. According our test (Fig. R3), when heating (70℃) was on, there did have a peak lasting several minutes (5–6 min) and then deceasing to the normal levels in ambient air, which means a new balancing process has been established. Heated filters are not suggested here because it will promote the thermal decomposition of ammonium salt in the particulate matter accumulated at the filter.

[Figure]

Figure R3. New balance established in 5–6 min after inlet heated.

Also it would be interesting to know how the calibration was done with a standard gas cylinder. The experience of the reviewer has been that the addition of a dry calibrated reference $NH_3$ gas leads to a complete loss of the $NH_3$ signal in some of these OAICOS types of instruments (as the instrument uses the water line for holding the $NH_3$ signal). Did the authors observe this? Was a humidified calibration done. The details of this would be useful information for the readers.

Response:We use a zero-gas supplier, in which the water in humid air is condensed mostly by the air compressor, to dilute the standard $NH_3$ gas from a cylinder. The water content in the zero gas is still abundant for holding the $NH_3$ signal. We compared the calibration results from a standard gas cylinder (Beifen, Beijing, China) and from a permeation tube (Fine metrology S.r.l.s., Spadafora, Italy) and there showed a comparable result (Fig. R4). As a relatively fixed dilution flow, which is close to the flow demand of the analyzer, can be set using a cylinder gas with a broader range of calibration curve, we prefer to use the cylinder gas instead of $NH_3$ permeation tube to calibrate the analyzer. While, if the $NH_3$ permeation tube used, the expected span concentrations only obtained by changing the flow of the dilution flow.

[Figure]

Figure R4. A comparable calibration result from a standard gas cylinder and from a permeation tube

In order to understand the response of the instrument, it would be good to have a figure with some of the measured data before averaging is done, particularly during the case study of the precipitation event perhaps or during Spring when the ammonia concentrations are highest. What was the estimated response time of the instrument? It is true that the response time is faster than when going from low to high concentrations, but it would be useful for the authors to characterise that for this setup – it is the response of the sample lines as well as the instrument. From the long term dataset a precision and accuracy and LOD should be presented.

Response: We have a test in summer, 2018. The sample air was switched between room air and outside air. The room air was with less humidity because air condition was on. And the outside air was with high humidity. $NH_3$ levels in room and outside air also show a great difference. The result is showed in Fig. R5, which is plotted with 1-min average data. Under these extreme changes of $H_2O$, $NH_3$ exhibited a response less than 1 hour. The response time is faster when going from low to high concentrations than from high to low one. The test tells us that it's sound to present the $NH_3$ in hourly mean, although minute-average data might have a limitation.

[Figure]

Figure R5. The response of NH$_3$ and H$_2$O as the sample air was switched between room air and outside air.

Results and Discussion: Figure 2: Please move baseline to zero as it is hard to interpret the NH$_3$ concentrations at the low end. I would prefer the NH$_3$ to be on its own graphs so that the reader can easily look at the NH$_3$ data which is the primary focus of the paper.

Response: We modified the figure (Fig. R6).

[Figure]

Figure R6. Temporal variations in the hourly average NH$_3$ mixing ratios, temperatures (T) and relative humidity (RH) at the urban and suburban stations in Beijing. Continuous thick lines were

smoothed with 168 points (7 days) by using the Savitzky–Golay method.

P6-7: Soil: There is some discussion around soil emissions of ammonia. I think it would be useful for the authors to make clear that soil itself does not emit ammonia per se, ammonia emissions from soils or vegetation are due to either fertiliser applied to the soil or ammonia deposition and re-emission. It would also be good to frame the discussion on the acidic soils with the aqueous acid-base chemistry, of which there is quite a big literature.

Response: Agree the opinion.

Section 3.2 and Section 3.3 and Supplementary material: The discussion of the differences in the seasonal and diurnal variability of $NH_3$ and $H_2O$ is really interesting and highlights the importance of understanding the boundary layer height and dilution/dispersion processes driving ambient ammonia concentrations – which are very high for an altitude of 70-100m! Though outside of the scope of this paper, a detailed modelling study of the datasets could be very insightful. However because of the complexities and rather small concentration changes I think that both of these sections need to be more conservative about the changes/trends/drivers of changes in concentrations. I would suggest through out that they authors should highlight uncertainties in the analysis.

Response: Agreed. In this paper, we report and compare the measurement results of $NH_3$ at urban and suburban Beijing. We are surprised by their difference in diurnal changes because the distance between the two sites is only 32 km away, which difference indicates a difference in source and sink. Section 3.2 and Section 3.3 are mainly the statistical results.

In particular the use of linear correlations is difficult to justify. In the supplementary material the linear correlation between T and RH of the $NH_3$ are plotted and the correlations summarised in Section 3.3. I am a bit concerned about this simple approach. There is much evidence that the ammonia equilibrium in the environment is non-linear – specifically it is exponential with equilibrium concentrations doubling for $\sim 5^{\circ}C$ in thermodynamic equilibrium. So rather than start with linear correlations the exponential model should be tested first (as the best theoretical relationship which has basis in physical chemistry).

Response: The linear correlations are for the average diurnal changes, not for all the hourly data (see fig. R7). In Fig. 2, scattering points are divergent, and the exponential model are not the best

fitting. Similar result can be found in Chang et al. (2016). Your suggestion will be considered in further analysis on selected data.

Chang, Y., Zou, Z., Deng, C., Huang, K., Collett, J. L., Lin, J. and Zhuang, G.: The importance of vehicle emissions as a source of atmospheric ammonia in the megacity of Shanghai, Atmospheric Chemistry and Physics, 16(5), https://doi.org/10.5194/acp-16-3577-2016, 2016.

[Figure]

Figure R7. Scatter plots between NH$_3$ and T/RH in different seasons.

Another concern was that the relationship was been studied over very small concentration ranges (<2ppb in some cases). Once the authors have assessed the precision and accuracy of their dataset, then error bars can be applied to these plots and then in some cases no line should be plotted, or a larger data group analysed. A non-linear relationship can be seen in some of the graphs. The opposite relationship is seen in a couple of plots, therefore it would be useful for the authors to look at those in a bit more detail. I would suggest a review of this section to make more clearly justifiable statistical analyses between ammonia concentration, relative humidity and temperature.

Response: Yes, for seasonal data, the error bars show a similar pattern with the average data (Fig.

R8). In the section, we statistically analyze the relationships between the average diurnal ammonia concentration with relative humidity and temperature. Yes, there showed small concentration ranges in the diurnal variation in a seasonal average, but for the hourly data, they exhibit more fluctuate (Fig. 1).

[Figure]

Figure R8. The average diurnal variations in $NH_3$ in different seasons.

Minor point: Some language checking would be useful.

Response: The paper has been edited in language by Wallace Academic Editing. We try the best to check the language again.

**Response to comments by referee 2**

**Anonymous Referee #2**

General comments: This manuscript reports a year long parallel measurements of $NH_3$ concentration at urban of suburban sites of Beijing. Statistical analysis, seasonal variation, diel profile and the relationship with temperature, RH, $H_2O$, wind speed and wind direct are present. This could be useful documentary for scientists who are interesting in aerosol chemistry, aerosol pH and pollutant management etc. I thus recommend publication of this manuscript on Atmospheric Chemistry and Physics.

Specific comments: 1. The author might be able to take advantage of the parallel measurements and summarize the difference in pattern of $NH_3$ pollution and source. However, there seems to be a lack of effort on the in-depth analysis.

Response: Thanks for the suggestions. We present the differences in the seasonal and diurnal variability of $NH_3$ and $H_2O$ at urban and suburban Beijing in this manuscript. The present spatial resolution of $NH_3$ emission inventory is not sufficient to support source difference analysis. Many other data should be used together to understand the $NH_3$ pollution and source, and this will be carried out in further studies.

2. The dew and $NH_3$ releasing point is interesting. I wonder if the dew point calculation could be useful to further explore and validate this point.

Response: Here, we reference some possible explanations to the diurnal changes. We calculate the dew points in summer for urban site (Fig. R9). There has some relationship but not all the truth.

[Figure]

Figure R9. The average diurnal change in $NH_3$, dew point, and $H_2O$ in summer at the urban Beijing.

3. The author claimed a rain clearing effect for $NH_3$. But in Figure 8 suburban case, there was an increasing during the rain and after the rain. Therefore, there is a inconsistence between the data and the explanation.

Response: We notice the phenomenon that there was a little increasing during the rain, and on average it was lower than the mean $NH_3$ concentration in the same time in August. We pointed that the diurnal variation of $NH_3$ on the rain day did not differ considerably from the average diurnal variation in August. At the suburban site, the diurnal $NH_3$ mixing ratio increased in the daytime, so there was an increasing after the rain. We pointed that the rainfall MIGHT have a clearing effect on $NH_3$ but needed more cases to support.

---

## Author Response (AR2)

***Response to Interactive comments on*** **"Measurement report: Exploring the NH₃ behaviours at urban and suburban Beijing: Comparison and implications" by Lan *et al*.**

Ziru Lan[1], Weili Lin[1], Weiwei Pu[2], and Ziqiang Ma[2,3]

[1]College of Life and Environmental Sciences, Minzu University of China, Beijing 100081

[2]Environmental Meteorological Forecast Center of Beijing-Tianjin-Hebei, Beijing, 100089, China

[3]Beijing Shangdianzi Regional Atmosphere Watch Station, Beijing, 101507, China

Comments to the Author:

The outstanding reviewer is satisfied that the technical issues have been addressed, however despite this being noted in the first round of reviews, I do not deem the quality of English to be of a sufficient standard for publication in ACP. Before publication, I ask the authors revise this for language, using the help of a fluent English speaker if necessary

Response:We thank both referees and editor for their constructive comments and suggestions. We revised the manuscript under the help of a fluent English speaker for language and modified some texts. The changes can be tracked in the marked-up version file.